# Overexpression of Grapevine *VvIAA18* Gene Enhanced Salt Tolerance in Tobacco

**DOI:** 10.3390/ijms21041323

**Published:** 2020-02-15

**Authors:** Wei Li, Changxi Dang, Yuxiu Ye, Zunxin Wang, Laibao Hu, Fan Zhang, Yang Zhang, Xingzhi Qian, Jiabin Shi, Yanyun Guo, Qing Zhou, Tailin Wang, Xinhong Chen, Feibing Wang

**Affiliations:** 1School of Life Science and Food Engineering, Huaiyin Institute of Technology, Huai’an 223003, China; liwei199933@163.com (W.L.); dcx_hnp@163.com (C.D.); mongye4597@163.com (Y.Y.); wangzunxin2020@163.com (Z.W.); hulaibao888@163.com (L.H.); hyitzy@163.com (Y.Z.); qianxz990108@163.com (X.Q.); jiabin19990421@163.com (J.S.); guoyy333@163.com (Y.G.); hgkjcz@hyit.edu.cn (Q.Z.); wtlxj@126.com (T.W.); 2Institute of Botany, Jiangsu Province and Chinese Academy of Sciences, Nanjing 210014, China; mumizhongfeng@126.com

**Keywords:** *Escherichia coli*, grapevine, salt tolerance, tobacco, *VvIAA18*

## Abstract

In plants, auxin/indoleacetic acid (Aux/IAA) proteins are transcriptional regulators that regulate developmental process and responses to phytohormones and stress treatments. However, the regulatory functions of the *Vitis vinifera* L. (grapevine) Aux/IAA transcription factor gene *VvIAA18* have not been reported. In this study, the *VvIAA18* gene was successfully cloned from grapevine. Subcellular localization analysis in onion epidermal cells indicated that VvIAA18 was localized to the nucleus. Expression analysis in yeast showed that the full length of *VvIAA18* exhibited transcriptional activation. Salt tolerance in transgenic tobacco plants and *Escherichia. coli* was significantly enhanced by *VvIAA18* overexpression. Real-time quantitative PCR analysis showed that overexpression of *VvIAA18* up-regulated the salt stress-responsive genes, including pyrroline-5-carboxylate synthase (*NtP5CS*), late embryogenesis abundant protein (*NtLEA5*), superoxide dismutase (*NtSOD*), and peroxidase (*NtPOD*) genes, under salt stress. Enzymatic analyses found that the transgenic plants had higher SOD and POD activities under salt stress. Meanwhile, component analysis showed that the content of proline in transgenic plants increased significantly, while the content of hydrogen peroxide (H_2_O_2_) and malondialdehyde (MDA) decreased significantly. Based on the above results, the *VvIAA18* gene is related to improving the salt tolerance of transgenic tobacco plants. The *VvIAA18* gene has the potential to be applied to enhance plant tolerance to abiotic stress.

## 1. Introduction

Salt and drought stresses seriously affect the productivity and cultivation scale of global crops, which are becoming serious threats to agricultural efforts to provide and satisfy the needs of a rapidly growing global population [1,2,3,4,5,6,7]. It has been reported that approximately 20% of the irrigated soils worldwide are suffering from salt stress [8]. Meanwhile, global water shortages and global climate change are threatening sustainable crop cultivation [9]. Based on increasingly severe environmental conditions, it is urgent to develop plants with high tolerance to salt and drought.

It has been reported that some plants have evolved counter-mechanisms, such as growth and development regulation, detoxification, ion homeostasis, and osmotic adjustment in order to adapt to the severe stresses environment (i.e., salinity, drought, and osmotic stress) [10,11]. Auxin plays a very important role in a wide variety of plant developmental and physiological processes [12,13]. It has been shown that response required for auxin execution is achieved through signaling, recruiting specific transcription factors to regulate the expression of down-stream genes [14]. Auxin/indoleacetic acid (Aux/IAA) proteins and auxin response factors (ARF) are two important protein families in development. They respond to phytohormones and stress treatment by controlling auxin-responsive transcription [13,15,16,17]. Although Aux/IAA proteins do not bind to auxin-responsive elements (AuxREs) directly, they control ARFs activity through protein protein interactions to regulate auxin-mediated gene expression [18,19]. ARFs protein binds to AuxREs in the promoter through the DNA binding domain, thereby regulating the expression of auxin-responsive genes [19,20,21].

To determine the molecular mechanism of auxin signaling, several Aux/IAA-encoding genes from different planned species, such as mung bean [22], *Arabidopsis* [23], rice [24,25] and grapevine [26], have been identified and characterized. There were 29 Aux/IAA gene family members in *Arabidopsis* [27], 31 members in rice [28], 26 members in the sorghum genome [29] and 26 members in grapevine [26]. In a variety of plant species, auxin is able to included many members of the Aux/IAA gene family to reponse [24,30,31,32]. Most *OsIAA* genes from rice were responsive to various abiotic stresses, indicating an interaction between plant growth and abiotic stresses [13]. The further study found that *OsIAA18* gene was induced by salt and drought stresses in rice [13]. However, cloning of grapevine *VvIAA* gene is rarely studied. Çakir et al. [26] screened the grapevine genome proteome, identified a total of 26 *VvIAA* genes showing high sequence identity, and isolated the *VvIAA4* gene. The further analysis found that *VvIAA4* expression was induced in response to 1-naphthaleneacetic acid (NAA) treatment, but was decreased by salt, drought, and salicylic acid (SA) treatments [26]. However, the regulatory functions of the grapevine Aux/IAA transcription factor gene *VvIAA18* in response to salt stress have not been reported.

In this study, we isolated the *VvIAA18* gene from grapevine and estimated its roles in transgenic tobacco. The heterologous overexpression of *VvIAA18* significantly enhanced salt tolerance in transgenic tobacco. Our results indicate that the *VvIAA18* gene is a promising option for enhancing the tolerance to abiotic stresses in plants.

## 2. Results

### 2.1. Cloning and Sequence Analysis of VvIAA18

The *VvIAA18* gene was cloned by reverse transcription PCR (RT-PCR). The ORF was 1191 bp in size and encoded a polypeptide that is 396 amino acids in length. The theoretical molecular weight of the protein was 43.61 kDa, and the theoretical isoelectric point (*p*I) was 8.33. At amino acid, residues 363-390, identified a putative nuclear localization signal sequence (Appendix A). The VvIAA18 protein contained an Aux/IAA protein domain (Appendix A) by/through sequence analysis via the InterProScan program (http://www.ebi.ac.uk/Tools/pfa/iprscan/) (Appendix A). 

A BLAST search results show that the amino acid sequence of VvIAA18 has higher amino acid identity with the protein product predicted by *Prunus mume* (XP_008228419, 64.52%), *Prunus persica* (XP_007215591, 63.91%), *Juglans regia* (XP_018843275, 58.31%), *Manihot esculenta* (OAY53082, 57.71%), *Gossypium arboreum* (XP_017607126, 56.62%), *Ricinus communis* (EEF46379, 56.36%), *Glycine max* (XP_003546582, 54.59%), *Populus euphratica* (XP_011014932, 52.13%), *Spinacia oleracea* (KNA11453, 49.37%), *Solanum lycopersicum* (NP_001266088, 48.24%), *Solanum tuberosum* (XP_015162257, 47.36%), *Nicotiana tabacum* (XP_016459020, 45.34%) and *Arabidopsis thaliana* (AAM65282, 41.10%) (Appendix A). Phylogenetic analyses revealed that VvIAA18 had closely related with the predicted protein products of *P. mume* (XP_008228419) and *P. persica* (XP_007215591) (Figure 1).

### 2.2. Nuclear Localization of VvIAA18

To provide further evidence for the potential role of VvIAA18 in transcriptional regulation, we performed a localization analysis of VvIAA18 subcellular in onion epidermal cells. The GFP fluorescence of VvIAA18-GFP was exclusively located in the nuclei of the cells, while the GFP control was distributed throughout the whole cell both in onion cells (Figure 2). These results indicate that VvIAA18 is a nuclear-localized protein.

### 2.3. Transcriptional Activation of VvIAA18

Usaully through VvIAA18was fused in-frame to GAL4 DNA-binding domain in the pGBKT7 vector and the fusion constructs pBD-VvIAA18 were transformed into the yeast strain AH109 to test whether VvIAA18 has transcription activity. The test results of yeast transformants harbouring pGAL4 and pBD-VvIAA18 grew normally on SD/Trp^−^/His^−^ medium exclusively and exhibited the activity of β-galactosidase reporter gene upon addition of X-gal on Whatman filter paper (Figure 3). Therefore, these results confirm that VvIAA18 is a transcription activator.

### 2.4. Improved Salt Tolerance in Escherichia coli

To examine the potential role of *VvIAA18* in protecting cells from salt stress, heterologous expression of *VvIAA18* in *Escherichia coli* (*E. coli*) (BL21) was carried out. The control was cells transformed with the empty vector. It is non-significant that the growth of the cells transformed either with empty vector or with recombinant plasmid, on fresh LB media. On solid media containing 0.5 M NaCl, the transformants expressing GST-VvIAA18 fusion protein showed higher growth rate than those expressed GST protein only (Figure 4a). On liquid media with 0.5 M NaCl, the growth rate of the transformants expressing GST-VvIAA18 fusion protein was threefold higher than the control after incubation for 10 h (Figure 4b). These results clearly indicate that heterologous expression of VvIAA18 protein enhanced the tolerance to salt stress of *E*. *coli*.

### 2.5. Regeneration and Identification of Transgenic Tobacco with VvIAA18 Gene

The ORF of *VvIAA18* was ectopically expressed in tobacco in the binary vector pCAMBIA1301-VvIAA18 (Figure 5a). The Wisconsin 38 tobacco varieties co-cultured with A.tumefaciens had a total of 250 leaf discs, and 17 putative transgenic plants were produced from 17 leaf discs. GUS assay showed that 6 of them had visible GUS activity in leaf, stem and root tissues, indicating stable *gusA* gene integration into the genome of the plants (Figure 5a–c). 

GUS expression was not observed in the remaining 11 WT strains (Figure 5a–c). Six independent transgenic lines overexpressing *VvIAA18* obtained by Hyg resistance selection were named L1-L6. PCR analysis further confirmed that these 6plants were transgenic (Figure 5e).

### 2.6. Enhanced Salt Tolerance in Tobacco Expressing VvIAA18

qRT-PCR analysis showed that the expression level of *VvIAA18* was significantly higher, from qRT-PCR analysis, in the transgenic lines, especially L2, L4, and L5, while no transgene expression was observed in WT (Figure 6). Therefore, transgenic lines L2, L4, and L5 were selected to assay further experiments.

The 3 transgenic lines (L2, L4, and L5) and WT were cultured on MS medium supplemented with no stress and 200 mM NaCl for 4 weeks, respectively. The transgenic plants exhibited significantly higher fresh weights in contrast to the poor-growing WT under salt stress, while no differences in growth and rooting were observed between the transgenic plants and WT under normal condition (Figure 7).

To further assess salt tolerance, the three transgenic plants and WT were planted in pots and irrigated with 200 mL of 200 mM NaCl solution for 8 weeks every 8 days. No significant difference was observed in the growth between the transgenic plants and the stress-free wild type. After 8 weeks of 200 mM NaCl stress, the transgenic plants showed good growth and increased physical size, while WT died (Figure 8). Fresh weight (FW) and dry weight (DW) of the 3 salt-tolerant plants were increased by 237–397% and 35–72%, respectively, compared to WT (Figure 8). These results demonstrated that the transgenic plants had enhanced salt tolerance compared to WT.

### 2.7. Southern Blot Analysis of Transgenic Plants

Southern blot analysis indicated that 3 salt-tolerant transgenic plants (lines L2, L4, and L5) displayed different integration patterns. The copy number of integrated *hpt*Ⅱ gene varied from 1 to 2, while WT had no hybridizing band as expected (Figure 9). Therefore, there is no obvious relationship between expression levels of related genes and the copy number of integrated *hpt*Ⅱ gene (Figure 9). Also, no clear relationship between the salt tolerance and the copy number was ascertained (Figure 9).

### 2.8. Increased Expression of the Salt Stress-Responsive Genes in Transgenic Plants

To dissect how overexpression of *VvIAA18* enhanced salt tolerance in transgenic tobacco palnts, the transcript levels of the 4 salt stress-responsive genes in the transgenic (L2, L4 and L5) and WT plants were examined by qRT-PCR. Under salt stress, well-known salt stress-responsive genes encoding pyrroline-5-carboxylate synthase (*NtP5CS*), late embryogenesis abundant protein (*NtLEA5*), *NtSOD* and *NtPOD* exhibited significantly increased expression levels in the transgenic plants compared with WT under 200 mM NaCl stress (Figure 10). The results indicated that *VvIAA18* might be involved in multiple regulatory pathways.

### 2.9. Promoted Stress Response Physiological Traits in Transgenic Plants

In plant research, proline, H_2_O_2_, MDA, SOD and POD, as important physiological indices, have been widely used to evaluate the plant stress responses [3,4,5,6,33]. Proline, H_2_O_2_ and MDA content and SOD and POD activities in the transgenic tobacco plants were measured by WT culture on stress and 200 mM NaCl MS medium for 4 weeks. These results indicated that proline content and SOD and POD activities were significantly increased, while H_2_O_2_ and MDA content were significantly decreased in the resistant plants compared to WT under salt stress (Figure 11). These results indicate that overexpression of *VvIAA18* inhibited ROS damage by decreasing H_2_O_2_ and MDA levels and enhancing antioxidant enzyme activities under salt stress.

## 3. Discussion

Soil salinity is one of the major factors that limit the productivity and quality of crops. Plant genetic engineering provides a promising method for breeding salt-tolerant varieties. An important strategy for improving salt tolerance of plants overexpression of salt tolerance related genes. 

In rice, the *OsIAA* genes play important roles in the developmental process and responses to phytohormones and stresses treatments [13,34]. The study found that the *OsIAA18* gene was induced by salt and drought stress in rice [13]. However, the regulatory functions of the grapevine *VvIAA18* in response to salt stress have not been reported. In this paper, we cloned the *VvIAA18* gene from grapevine. Sequence analysis showed that VvIAA18 protein contained an Aux/IAA protein domain (Appendix A). The VvIAA18 protein was localized in the nucleus (Figure 2). Further transcriptional activation analysis found that the VvIAA18 protein was a transcription activator (Figure 3). Expression of *VvIAA18* enhanced osmotic salt stress tolerance in *E. coli* (Figure 4). Constitutive overexpression of *VvIAA18* also significantly enhanced salt tolerance in transgenic tobacco plants (Figure 7 and Figure 8). Thus, it is thought that the *VvIAA18* gene may play an important role in response to salt stress in plants.

Osmotic stress often leads to more proline accumulation, and the level of proline accumulation is related to the extent of salt tolerance [3,4,5,6,7]. In this study, the transgenic tobacco plants had significantly higher proline content compared to WT under salt stress, indicating measurable improvement of salt tolerance (Figure 11). Proline accumulation in the *VvIAA18*-overexpressing tobacco plants most likely maintains the osmotic balance between the intracellular and extracellular environment under salt stress, which results in the improved salt tolerance [7,35]. Also, proline helps cells to maintain membrane integrity [7,35,36] and has been proposed to function as molecular chaperone stabilizing the structure of proteins [37]. Therefore, it is assumed that proline accumulation in the *VvIAA18*-overexpressing tobacco plants might protect the cell membrane from salt-induced injuries. It was also found that the expression of *NtP5CS* gene was up-regulated in the transgenic plants under salt stress (Figure 10). Thus, the present results suggest that up-regulating the expression of the *NtP5CS* gene in transgenic tobacco plants to increase the accumulation of proline will cause *VvIAA18* overexpression. (Figure 11).

Plant cells ROS induced by salt stress. It is important to maintain a stronger ROS-scavenging ability under salt stress to alleviate the induced oxidative damage, especially in plant leaves where photosynthesis is dramatically impacted [38]. SOD is the first line of defense against ROS induced by salt to promote the conversion of superoxide into oxygen and H_2_O_2_, which is further scavenged by coordinated action of POD etc. [3,4,35,39]. In this study, H_2_O_2_ accumulation in tobacco plants *VvIAA18*-overexpressing was significantly lower than WT under salt stress (Figure 10). A systemic up-regulation of ROS scavenging genes (*NtSOD* and *NtPOD*) and significantly increase of antioxidant enzyme (SOD and POD) activities were observed in the transgenic plants under salt stress (Figure 10 and Figure 11). Therefore, the improved salt tolerance of the transgenic plants might be due to at least in partially enhanced ROS scavenging capacity [35,36,40,41]. In addition, proline is an effective scavenger of singlet oxygen and hydroxyl radicals [7,35]. Our results support that more proline accumulation activates ROS scavenging system, leading to the enhanced salt and drought tolerance in the *VvIAA18*-overexpressing tobacco plants (Figure 12) [35,36,39,41]. 

In plants, MDA is a marker for lipid peroxidation [35,36,39,41]. Higher levels of MDA can induce cell membrane damage and further reduces abiotic stresses tolerance of plants [42,43]. The present study found that MDA content was significantly lower in the *VvIAA18*-overexpressing tobacco plants than in WT under salt stress, suggesting that transgenic plants have stronger salt tolerance (Figure 11).

It has been reported that LEA proteins play pivotal roles in stress tolerance, as osmotic adjustment material and a protection material for cell membrane structure [44,45]. The *LEA* genes have been shown to enhance plant tolerance to salt stress [3,5]. In this study, the significant increase in salt tolerance of tobacco plants *VvIAA18*-overexpressing under salt stress was reflected by the upregulation of *NtLEA5* (Figure 11 and Figure 12).

## 4. Materials and Methods

### 4.1. Plant Mterials

The *VvIAA18* gene was cloned using grapevine cultivar PN40024 in this study. The expressed sequence tag (EST) of *VvIAA18* was obtained from the cDNA-AFLP data of PN40024. Tobacco (*Nicotiana tabacum* L.) cultivar Wisconsin 38 [wild type (WT)], as a model plant, was used to study the functions of *VvIAA18*.

### 4.2. Cloning of Grapevine VvIAA18 Gene

Total RNA was extracted from freshly leaves of PN40024 with the RNAprep Pure Kit (Tiangen Biotech, Beijing, China). RNA samples were reverse-transcribed according to the instructions of Quantscript Reverse Transcriptase Kit (Tiangen Biotech, Beijing, China). Based on the sequence of *VvIAA18* (Genbank accession No. XM_010656314), we designed one gene-specific primers (GC-F/R) of RT-PCR (Appendix A) to obtain its full-length cDNA sequence. PCR was performed with an initial denaturation 94 °C for 3 min, followed by 35 cycles of 94 °C for 30 s, 55 °C for 30 s, 72 °C for 1 min and final extension 72 °C for 10 min. PCR products were separated on a 1.0% (*w/v*) agarose gel. Target DNA bands were recovered by gel extraction, then cloned into PMD19-T (TaKaRa, Beijing, China), and finally transformed into competent cells of *E. coli* strain DH5α. Examine hite colonies were by PCR and sequence positive colonies (Invitrogen, Beijing, China).

### 4.3. Sequence Analysis of VvIAA18 Gene

The full-length cDNA of *VvIAA18* was analyzed by an online BLAST at the National Center for Biotechnology Information (NCBI) website (http://www.ncbi.nlm.nih.gov/). For the multiple sequence alignment analysis, the DNAMAN software (Lynnon Biosoft, Quebec, Canada) was used to align the amino acid sequences of *VvIAA18* and other IAA homologs from different plant species from NCBI. The phylogenetic analysis was conducted with the MEGA4 software (http://www.megasoftware.net/). The ProtParam tool (http://web.expasy.org/protparam/) was used to calculate the theoretical molecular weight and isoelectric point (*p*I). (http://web.expasy.org/protparam/). The conserved domain of VvIAA18 protein was scanned by InterProScan program (http://www.ebi.ac.uk/Tools/pfa/iprscan/). The cNLS Mapper program was used to predict the nuclear localization signal of VvIAA18 protein (http://nls-mapper.iab.keio.ac.jp/cgi-bin/NLS_Mapper_form.cgi).

### 4.4. Subcellular Localization and Transactivation Assay of VvIAA18

Subcellular localization of VvIAA18 in onion (*Allium cepa*) epidermal cells was analyzed as described by Wang et al. [4]. The ORF of *VvIAA18* was cloned and then inserted into the pMDC83 expressing vector containing the green fluorescent protein gene (GFP) at SpeⅠand AscⅠrestriction sites under the control of the CaMV35S promoter and NOS (nopaline synthase) terminator (Appendix A). Both the fusion construct (*VvIAA18*-GFP) and the control vector (GFP) were transformed into living onion epidermal cells by particle bombardment with a GeneGun (Biorad HeliosTM) according to the instruction manual (helium pressure 260 psi). After incubation on MS medium (pH 5.8) solidified with 3% agar at 28 °C for 24-36 h, the onion cells were observed with a bright field and fluorescence using confocal microscopy (Nikon Inc., Melville, NY). Transactivation assay of VvIAA18 in yeast (*Saccharomyces cerevisiae*) was conducted according to the method of Jiang et al. [46]. GUS activities were assayed by the fluorometric method [47].

### 4.5. Protein Expression and Bacterial Growth Analysis

Based on the method of Li et al. [39], the VvIAA18 fragment released by the *BamH* I and *EcoR* I digestions was cloned onto a pGEX-4T-1 expression vector digested with the same enzymes (NEB, USA). The resulting pGST-VvIAA18 plasmid was transformed into *Escherichia coli* (BL21) for protein expression. The expression of the GST-VvIAA18 fusion protein was induced with 0.2 mM isopropylb-D-thiogalactopyranoside (IPTG) for 3 h at 37 °C. The cell cultures were serially diluted (1:10, starting OD_600nm_ 1.0) before spotting on LB agar plates supplemented with or without 0.5 M NaCl and incubated at 37 °C for 24 h. For growth analysis in liquid medium, 100 µL of cell culture (OD_600nm_ 1.0) was inoculated into 10 mL LB medium supplemented with or without 0.5 NaCl. Cell growth densities were measured at 600nm.

### 4.6. Transformation of Tobacco with VvIAA18

The binary vector pCAMBIA1301-*VvIAA18* used in this study contained the *VvIAA18* gene under the control of the cauliflower mosaic virus (CaMV) 35S promoter and the nopaline synthase (NOS) terminator and β-glucuronidase (*gusA*) and hygromycin resistance (*hpt*Ⅱ) genes driven by a CaMV 35S promoter, respectively. Primer5 was used to design primers for amplification of *VvIAA18* (Appendix A). The vector pCAMBIA1301-*VvIAA18* was transformed into the *Agrobacterium tumefaciens* strain EHA105 cells by the electroporation method for tobacco transformation [48]. Transformation and plant regeneration were performed according to the method of Jiang et al. [49].

### 4.7. Molecular Confirmation of Transgenic Plants

The putatively transgenic tobacco plants were identified using the histochemical GUS assay according to Jefferson et al. [47]. The blue staining of tissues indicated a positive reaction. Genomic DNA was extracted from the leaves of the GUS-positive plants, and PCR amplifications were performed using specific primers (Appendix A) to amplify fragments of the *hpt*Ⅱ coding sequence.

### 4.8. Assay for Salt Tolerance

In vitro assay for salt tolerance was conducted as described by Jiang et al. [46]. Transgenic tobacco and WT plants were cultured on MS medium with 200 mM NaCl in order to evaluate their in vitro salt tolerance at (27±1) °C under 13 h of cool-white fluorescent light at 3 000 Lux. The growth and rooting ability were continuously observed for 4 weeks, and then their fresh weights were measured.

In vivo assay for salt tolerance was based on the method of Wang et al. [4]. Transgenic and WT tobacco plants were grown in 19-cm diameter pots containing a mixture of soil, vermiculite, and humus (1:1:1, *v/v/v*) in a greenhouse, with nine plants per pot. All pots were irrigated sufficiently with half-Hoagland solution for 2 weeks under normal condition. Each pot was then irrigated with a 200 mL of 200 mM NaCl solution once every 2 days for 8 weeks. After treatment, the plant FW was measured immediately. The plants were then dried for 24 h in an oven at 80 °C and weighed DW. All treatments were performed in triplicate.

### 4.9. Southern Blot Analysis

The cetyltrimethylammonium bromide (CTAB) method was used that Genomic DNA was extracted from the leaves of transgenic and WT plants [50]. Southern blot analysis was conducted as described by Wang et al. [5]. Coding sequence of the 591 bp *hpt*Ⅱwas used as probe (Appendix A). The labeling of probe, prehybridization, hybridization and detection were performed using DIG High Prime DNA Labeling and Detection Starter Kit II (Roche Diagnostics GmbH, Germany).

### 4.10. Expression Analysis of VvIAA18 and the Related Genes

The expression of *VvIAA18* gene and the salt stress-responsive genes was analyzed by real-time quantitative PCR (qRT-PCR) as described by Wang et al. [4]. Transgenic tobacco and WT plants were cultured for 4 weeks on MS medium with no stress and 200 mM NaCl. The cDNA solution was used as templates for PCR amplification with gene specific primers (Appendix A). Tobacco *Ntactin* gene was used as an internal control [49] (Appendix A). Quantification of gene expression was done with the comparative *C*_T_ method [51]. All experiments were repeated three times and each data represents the average of three experiments.

### 4.11. Analyses of Proline, H2O2, and MDA Content and SOD and POD Activities

The content of proline, H_2_O_2_ and malonaldehyde (MDA) and the activities of superoxide dismutase (SOD) and peroxidase (POD) in the transgenic tobacco plants and WT cultured for 4 weeks on a MS medium with no stress and 200 mM NaCl were measured according to the method of Wang et al. [4].

### 4.12. Statistical Analysis

The experiments were repeated three times and the data presented as the mean ± standard error (SE). Where applicable, data were analyzed by Student’s *t*-test in a two-tailed analysis. *p* < 0.05 or *p <* 0.01 was considered to be statistically significant.

## 5. Conclusions

The *VvIAA18* gene has been successfully cloned from grapevine. Overexpression of *VvIAA18* significantly enhanced salt tolerance in transgenic tobacco plants. Our results suggest that the enhanced salt tolerance in the *VvIAA18*-overexpressing tobacco plants is due to altering salt tolerance associated components by up-regulating salt stress-responsive genes. The *VvIAA18* gene is a hopeful candidate for enhancing tolerance to abiotic stresses in plants.

## Figures and Tables

**Figure 1 ijms-21-01323-f001:**
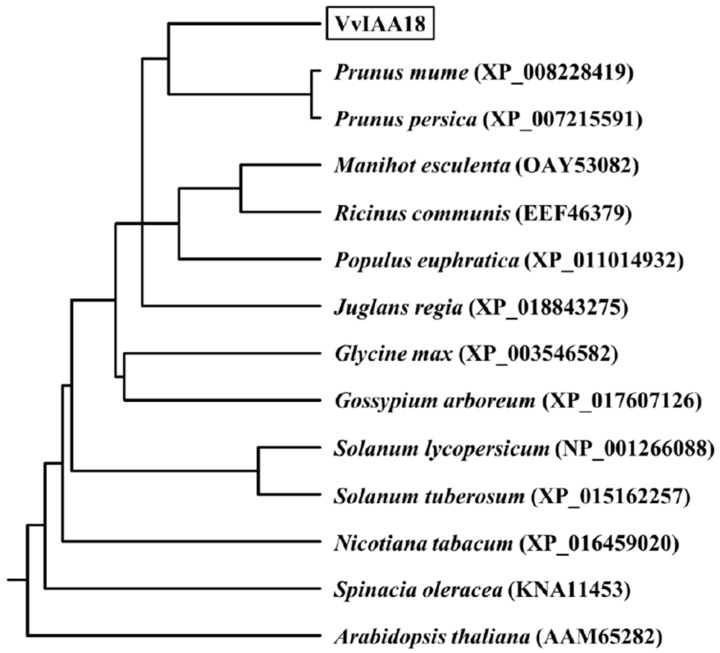
A phylogenetic tree of the VvIAA18 protein with its other plant homologous proteins. The branch lengths are proportional to the distance.

**Figure 2 ijms-21-01323-f002:**
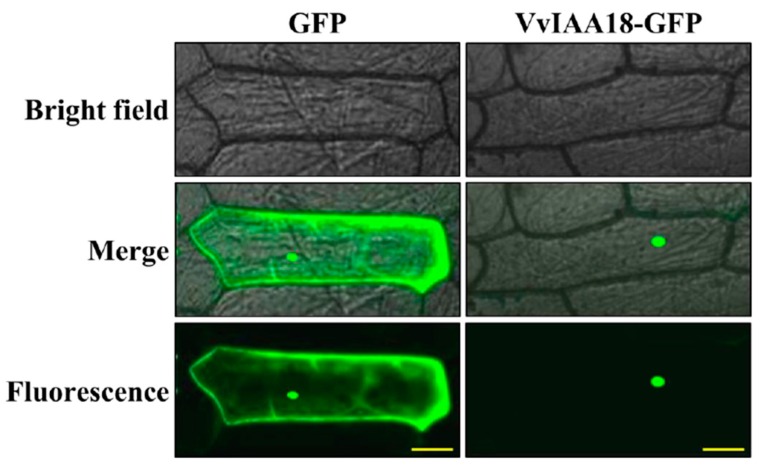
Subcellular localization of the VvIAA18 protein in onion epidermal cells. The VvIAA18-GFP fusion protein was localized to the nucleus. Scale bar = 100 μm.

**Figure 3 ijms-21-01323-f003:**
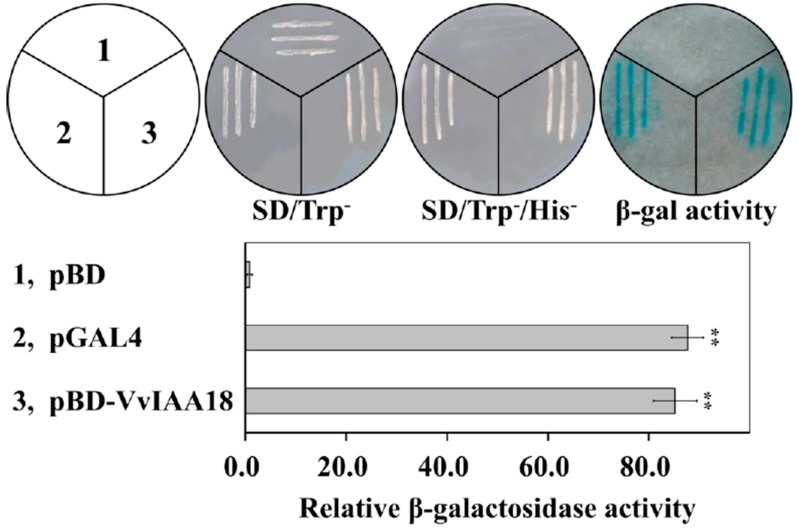
Transactivation assay of the VvIAA18 protein in the yeast. Fusion protein of the GAL4 DNA-binding domain and VvIAA18 were expressed in yeast. Use the empty pBD (pGBKT7) vector (negative control) and the pGAL4 vector (positive control). The culture solution of the transformed yeast was dropped onto SD plates without tryptophan or histidine. The plates were incubated for 3 days and then subjected to β-galactosidase assay. The results were analyzed by Student’s *t*-test in a two-tailed analysis. Significance was defined as *p* < 0.01 (**).

**Figure 4 ijms-21-01323-f004:**
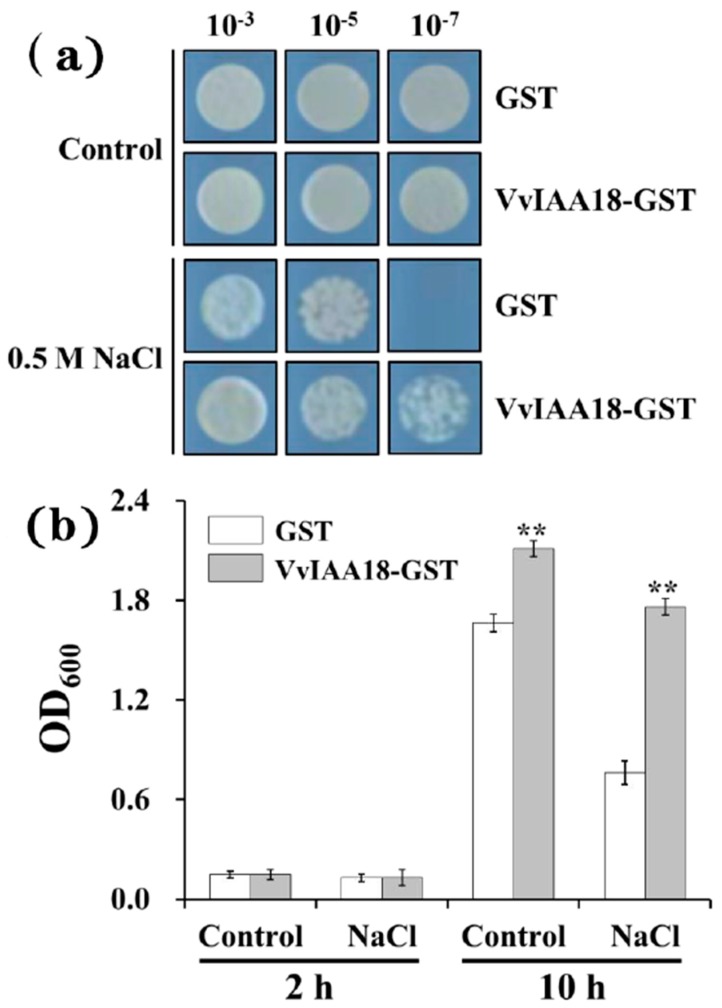
Enhanced salt tolerance in *Escherichia coli*. (**a**) Growth analysis of cells spotted on LB agar plate supplemented with 0.5 M NaCl. (**b**) Growth analysis of cells cultured in liquid medium supplemented with 0.5 M NaCl. Cell growth densities were measured at 600 nm at the indicated time points. The results were analyzed by Student’s *t*-test in a two-tailed analysis. Significance was defined as *p* < 0.01 (**).

**Figure 5 ijms-21-01323-f005:**
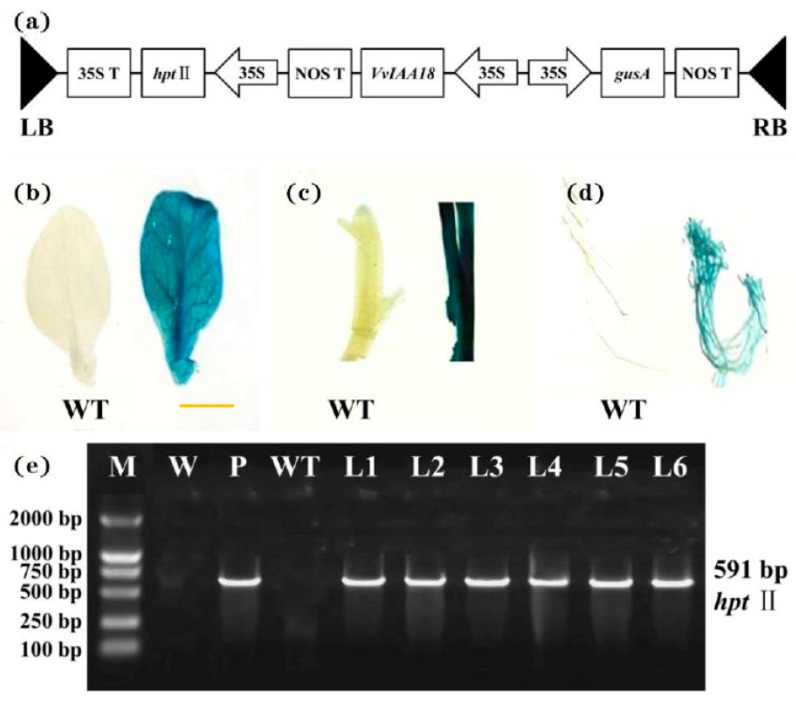
Molecular confirmation of transgenic plants. (**a**) Schematic diagram of the double plasmid pCAMBIA1301-*VvIAA18* T-DNA region. LB, left border; RB, right border; *hpt*Ⅱ, hygromycin phosphotransferase Ⅱ gene; *VvIAA18*, grape Aux/IAA transcription factor gene; *gusA*, β-glucuronidase gene; 35S, cauliflower mosaic virus (CaMV) 35S promoter; 35S T, CaMV 35S terminator; NOS T, nopaline synthase terminator. (**b**–**d**) β-glucuronidase (GUS) expression in leaf, stem, and root of a transgenic plant and no GUS expression in the wild-type (WT) (bar = 10 mm). (**e**) PCR analysis of *VvIAA18*-overexpressing tobacco plants. Lane M, DL2000 DNA marker; Lane W, water as negative control; Lane P, plasmid pCAMBIA1301-*VvIAA18* as positive control; Lane WT, wild type; Lanes L1-L6, different transgenic lines.

**Figure 6 ijms-21-01323-f006:**
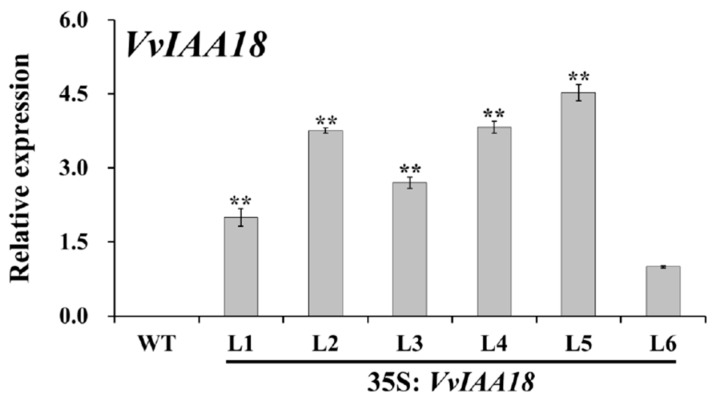
Real-time quantitative PCR expression analysis of *VvIAA18* gene in transgenic tobacco plants. The expression of *VvIAA18* in vitro-grown plants of 6 transgenic plants and wild type was analyzed. The tobacco *Ntactin* gene was used as an internal control. Data are presented as means ± SE (n = 3). The results were analyzed by Student’s *t*-test in a two-tailed analysis. Significance was defined as *p* < 0.01 (**).

**Figure 7 ijms-21-01323-f007:**
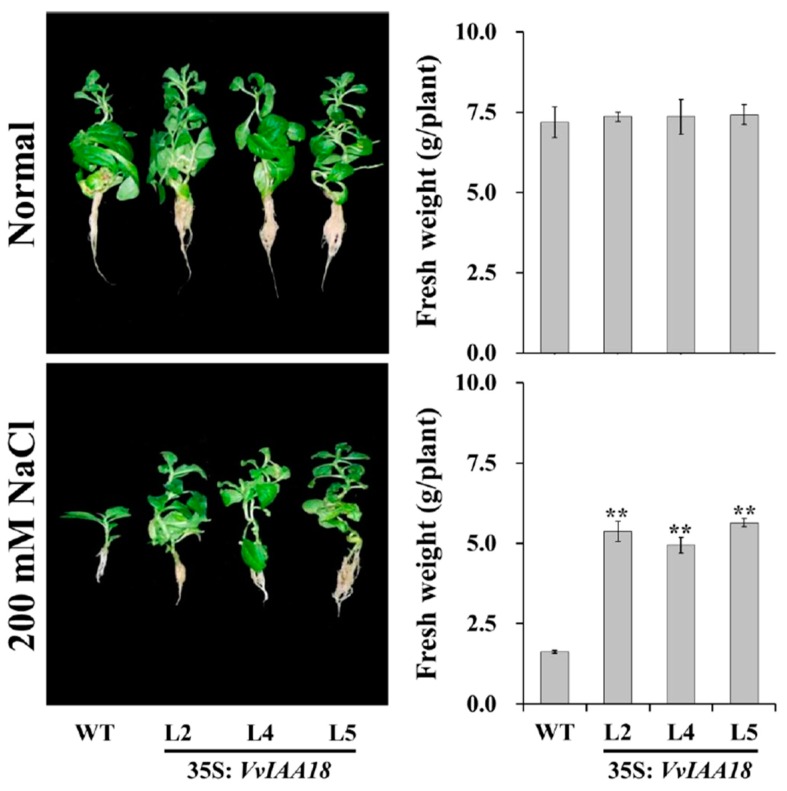
The growth and rooting of transgenic plants were compared and WT was cultured for 4 weeks on MS medium with no stress or 200 mM NaCl. Data are presented as means ± SE (n=3). The results were analyzed by Student’s *t*-test in a two-tailed analysis. Significance was defined as *p* < 0.01 (**).

**Figure 8 ijms-21-01323-f008:**
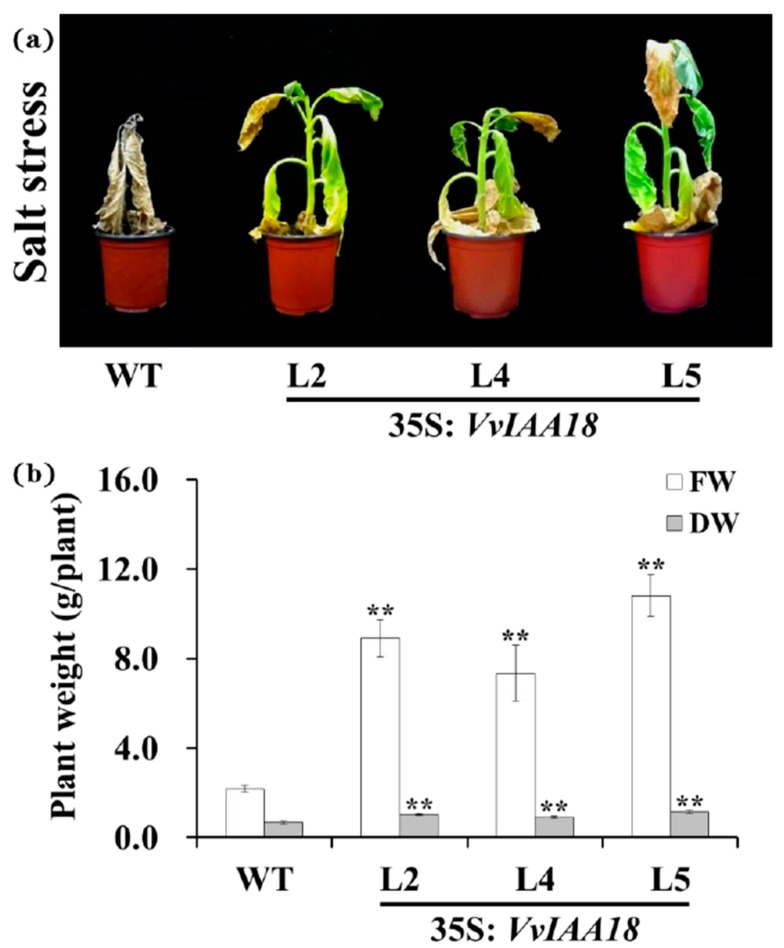
Responses of the transgenic tobacco plants and WT grown in pots under salt stress. (**a**), Phenotypes of the transgenic tobacco plants grown in pots under 200 mM NaCl stress. (**b**), Biomass of the transgenic tobacco plants grown in pots under 200 mM NaCl stress. Data are presented as means ± SE (n = 3). The results were analyzed by Student’s *t*-test in a two-tailed analysis. Significance was defined as *p* < 0.01 (**).

**Figure 9 ijms-21-01323-f009:**
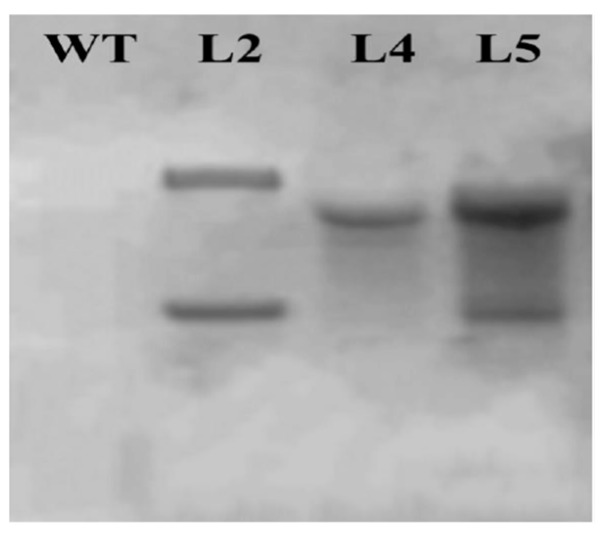
Southern blot analysis of the transgenic plants to detect the copy number of integrated *hpt*Ⅱ gene. WT, wild type; L1, L4 and L5, enhanced salt tolerance transgenic plants.

**Figure 10 ijms-21-01323-f010:**
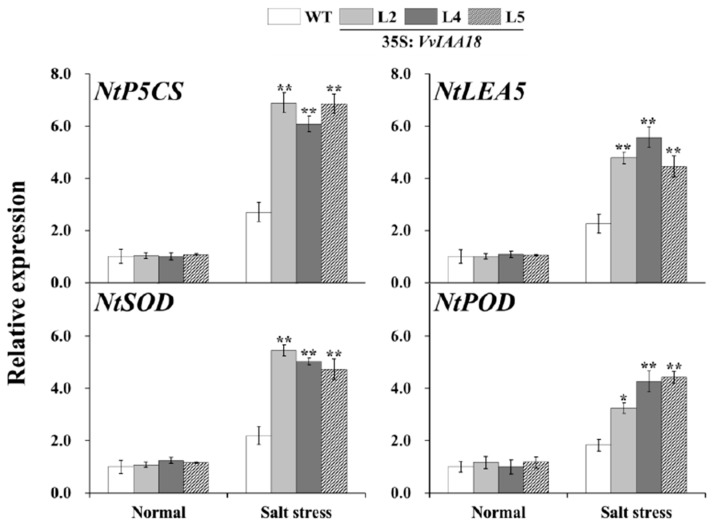
Relative expression level of salt stress-responsive genes in the leaves of transgenic tobacco plants and WT under salt stress. The tobacco *Ntactin* gene was used as an internal control. Results are expressed as relative values with respect to WT, which was set to 1.0. Data are presented as means ± SE (*n* = 3). The results were analyzed by Student’s *t*-test in a two-tailed analysis. Significance was defined as *p* < 0.05 and *p* < 0.01 (**).

**Figure 11 ijms-21-01323-f011:**
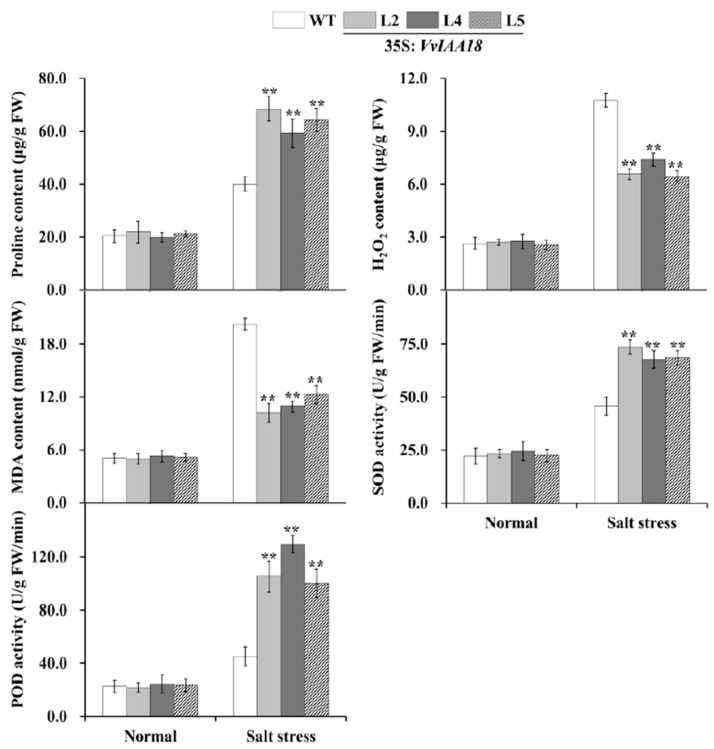
The content of proline, H_2_O_2_ and malondialdehyde(MDA), and the activities of superoxide dismutase (*NtSOD*) and peroxidase (*NtPOD*) in the leaves of transgenic tobacco plants and WT under salt stress. Data are presented as means ± SE (*n* = 3). The results were analyzed by Student’s *t*-test in a two-tailed analysis. Significance was defined as *p* < 0.01 (**).

**Figure 12 ijms-21-01323-f012:**
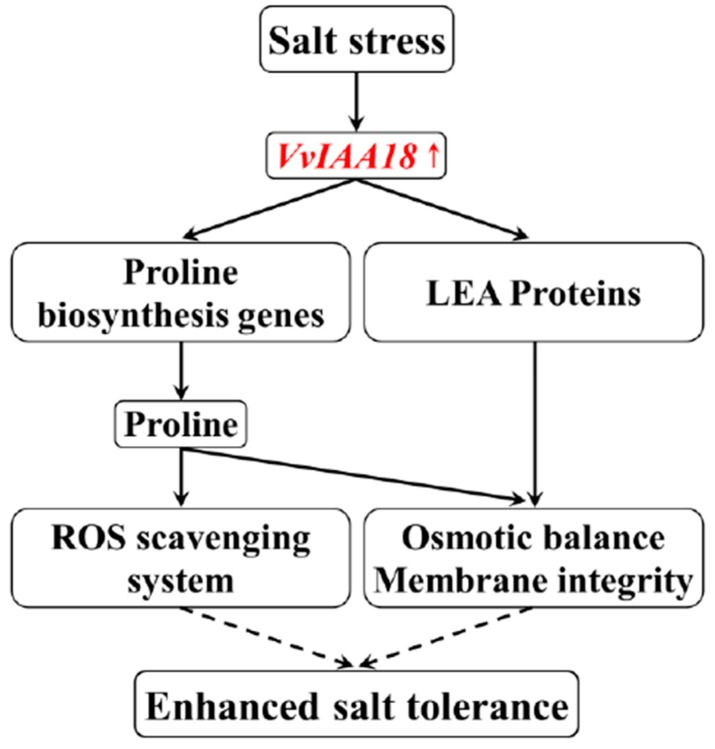
Hypothesis of the regulatory network of the *VvIAA18* gene involved in salt stress response. Constitutive expression of *VvIAA18* up-regulates the genes involved in proline biosynthesis and reactive oxygen species (ROS) scavenging, which result in significant physiological changes, including increased proline level and reduced ROS accumulation, leading to the enhanced salt tolerance.

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
