# Peer review of "Overexpression of Grapevine VvIAA18 Gene Enhanced Salt Tolerance in Tobacco"

_ijms, 2020, doi:10.3390/ijms21041323_

Round 1
Reviewer 1 Report
The MS by Li et al describes the cloning and functional analysis of grapevine IAA18 gene. The gene was cloned from grapevine and its function was tested by expression in yeast, E Coli and tobacco. The research work provides convinced evidence that Vv IAA is a transcription factor and overexpression can improve bacterial and plant tolerance to salt stress. The methods are correct, and the results were properly interpreted. The MS is well organized, but still needs English edits for better flow and easy understanding. Some other suggestions are listed below.
The title is not very accurate. I would suggest “overexpression of grapevine IAA18 gene enhanced salt tolerance in tobacco”
Latin name for grapevine was not provided
The rational for studying VvIAA18 was not provided? Why did authors still want to investigate this gene from grapevine since the homolog of rice gene has been shown to enhance salt tolerance? I would argue we can use OsIAA 18 to engineer other crops for salt tolerance? Why should I chose VvIAA18?
Results 2.1 is any intron for VvIAA 18? What is the NLS sequence? Figure 1 can be presented as supplementary data.
Figure 3, please explain the reason why GFP alone can enter nucleus since there is no NLS in GFP protein?
Figure 6b, c d needs label for transgenic plants in GUS staining of leaf, stem and roots
Figure 7. does Tobacco have IAA18 homologs? Were your qPCR primers specific to VvIAA 18 gene? Those primers were not provided in methods.
Figure 8. Were there any other symptom shown on wildtype plants when placed on medium with high concentrations of salt. I would expect some chlorosis symptom development. The photo shows the high concentration of salt only inhibit growth? it would be good to have more plants for wildtype and transgenic genotypes
Figure 10 Southern blot using hptII gene would be not suitable for detecting IAA18 insertion copies because IAA18 can disconnect from hptII gene during the insertion?
Some methods lack details, for example how did you make GFP and IAA-GFP fusion constructs?
Author Response
Dear Editor:
We greatly appreciate the efforts of you and the reviewer for the constructive comments that have helped shape this manuscript into a better form. We have addressed all the concerns by either editing the manuscript or clarifying the details. All changes made to the text are in red color. Please see the detailed point-by-point response added below.
Sincerely,
Best wishes!
Feibing Wang
E-mail: wangfeibing1986@163.com
Point by point response to Reviewer 1
The title is not very accurate. I would suggest “overexpression of grapevine IAA18 gene enhanced salt tolerance in tobacco”
Response: Thank you.
Latin name for grapevine was not provided
Response: Thank you. Revised based on comments. Please see Line 14.
The rational for studying VvIAA18 was not provided? Why did authors still want to investigate this gene from grapevine since the homolog of rice gene has been shown to enhance salt tolerance? I would argue we can use OsIAA18 to engineer other crops for salt tolerance? Why should I chose VvIAA18?
Response: Thank you. Revised based on comments. As described in Lines 280-281, ‘The expressed sequence tag (EST) for VvIAA18 was obtained from the cDNA-AFLP data of PN40024’.
Results 2.1 is any intron for VvIAA18? What is the NLS sequence? Figure 1 can be presented as supplementary data.
Response: Thank you. Revised based on comments. We cloned cDNA of grapevine VvIAA18 gene. Please see Lines 287-289. NLS sequence was nuclear localization signal. Please see Lines 304-305. Figure 1 was presented as supplementary data Figure s1.
Figure 3, please explain the reason why GFP alone can enter nucleus since there is no NLS in GFP protein?
Response: Thank you. Revised based on comments. In our study, GFP alone was in the nucleus, similar to the results reported by Wang et al. (2019, 2018), Huo et al. (2018) and Kang et al. (2018).
[1] Wang FB, Ren XQ, Zhang F, Qi MY, Zhao HY, Chen XH, Ye YX, Yang JY, Li SG, Zhang Y, Niu Y, Zhou Q. 2019. A R2R3-type MYB transcription factor gene from soybean, GmMYB12, is involved in flavonoids accumulation and abiotic stress tolerance in transgenic Arabidopsis. Plant Biotechnology Reports, 13: 219-233.
[2] Wang FB, Ren GL, Li FS, Wang BW, Yang YL, Ma XW, Niu Y, Ye YX, Chen XH, Fan S, Zhou Q. 2018. Overexpression of GmSnRK1, a soybean sucrose non-fermenting-1 related protein kinase 1 gene, results in directional alteration of carbohydrate metabolism in transgenic Arabidopsis. Biotechnology & Biotechnological Equipment, 32(4): 835-845.
[3] Huo JX , Du B, Sun SF, He SZ, Zhao N, Liu QC, Zhai H. A novel aldo-keto reductase gene, IbAKR, from sweet potato confers higher tolerance to cadmium stress in tobacco. Frontiers of Agricultural Science and Engineering, 2018, 5(2): 206-213.
[4] Kang C, Zhai H, He SZ, Zhao N, Liu QC. A novel sweetpotato bZIP transcription factor gene, IbbZIP1, is involved in salt and drought tolerance in transgenic Arabidopsis. Plant Cell Reports, 2019:1-10.
Figure 6b, c d needs label for transgenic plants in GUS staining of leaf, stem and roots
Response: Thank you. Revised based on comments. Please see Figure 5 b,c,d (bar = 10 mm).
Figure 7. does Tobacco have IAA18 homologs? Were your qPCR primers specific to VvIAA 18 gene? Those primers were not provided in methods.
Response: Thank you. Revised based on comments. Based on cDNA of VvIAA18, we designed gene specific primers of qRT-PCR, Please Table s1.
The expression of VvIAA18 gene and the salt stress-responsive genes was analyzed by real-time quantitative PCR (qRT-PCR) as described by Wang et al.
Response: Thank you. Revised based on comments. Well-known salt stress-responsive genes encoding NtP5CS, NtLEA5, NtSOD and NtPOD was analyzed by qRT-PCR, please see Figure 10.
Figure 8. Were there any other symptom shown on wildtype plants when placed on medium with high concentrations of salt. I would expect some chlorosis symptom development. The photo shows the high concentration of salt only inhibit growth? it would be good to have more plants for wildtype and transgenic genotypes
Response: Thank you. Revised based on comments. As described in Lines 151-154, ‘The transgenic plants exhibited significantly higher fresh weights in contrast to the poor-growing WT under salt stress, while no differences in growth and rooting were observed between the transgenic plants and WT under normal condition’.
Figure 10 Southern blot using hpt II gene would be not suitable for detecting IAA18 insertion copies because IAA18 can disconnect from hpt II gene during the insertion?
Response: Thank you. Revised based on comments. In Figure 9, southern blot analysis was conducted based on coding sequence of the 591 bp hptâ…¡, similar to the results reported by Wang et al. (2018) and Liu et al. (2014).
[1] Wang FB, Ren GL, Li FS, Qi ST, Xu Y, Wang BW, Yang YL, Ye YX, Zhou Q, Chen XH. 2018. A chalcone synthase gene AeCHS from Abelmoschus esculentus regulates flavonoids accumulation and abiotic stress tolerance in transgenic Arabidopsis. Acta Physiologiae Plantarum, 40:97.
[2] Liu DG, Wang LJ, Liu CL, Song XJ, He SZ, Zhai H, Liu QC. 2014. An Ipomoea batatas iron-sulfur cluster scaffold protein gene, IbNFU1, is involved in salt tolerance. PLoS ONE, 9, e93935.
Some methods lack details, for example how did you make GFP and IAA-GFP fusion constructs?
Response: Thank you. Revised based on comments. We add the details of subcellular localization of VvIAA18, Please see Lines 305-312.

Reviewer 2 Report
The paper is well written.
The paper is very well and clearly written, the experimental approach is well detailed, the statistical analysis of the obtained results is appropriate and confirm the study hypothesis. The paper has a real potential to be cited, taking into account the global problem of climate change, that could drastically affect the soil quality and plant tolerance to different abiotic stresses; therefore, research aiming to understand the molecular mechanisms of plant tolerance to different stress conditions is needed.Author Response
Dear Editor:
We greatly appreciate the efforts of you and the reviewer for the constructive comments that have helped shape this manuscript into a better form. We have addressed all the concerns by either editing the manuscript or clarifying the details. All changes made to the text are in red color. Please see the detailed point-by-point response added below.
Sincerely,
Best wishes!
Feibing Wang
E-mail: wangfeibing1986@163.com
Point by point response to Reviewer 2
The paper is well written. The paper is very well and clearly written, the experimental approach is well detailed, the statistical analysis of the obtained results is appropriate and confirm the study hypothesis. The paper has a real potential to be cited, taking into account the global problem of climate change, that could drastically affect the soil quality and plant tolerance to different abiotic stresses; therefore, research aiming to understand the molecular mechanisms of plant tolerance to different stress conditions is needed.
Response: Thank you.